# Oral Immunization with Recombinant *Saccharomyces cerevisiae* Expressing Viral Capsid Protein 2 of Infectious Bursal Disease Virus Induces Unique Specific Antibodies and Protective Immunity

**DOI:** 10.3390/vaccines11121849

**Published:** 2023-12-14

**Authors:** Huliang Li, Deping Hua, Qingxia Qu, Hongwei Cao, Zhehan Feng, Na Liu, Jinhai Huang, Lei Zhang

**Affiliations:** School of Life Sciences, Tianjin University, Tianjin 300072, China; 2020226031@tju.edu.cn (H.L.); huadeping@tju.edu.cn (D.H.); qqx539111@126.com (Q.Q.); honvvcao@163.com (H.C.); rd4944823xig@163.com (Z.F.); leaflana@163.com (N.L.)

**Keywords:** IBDV, VP2, oral immunization, vaccine

## Abstract

Infectious bursal disease (IBD), as a highly infectious immunosuppressive disease, causes severe economic losses in the poultry industry worldwide. *Saccharomyces cerevisiae* is an appealing vehicle used in oral vaccine formulations to safely and effectively deliver heterologous antigens. It can elicit systemic and mucosal responses. This study aims to explore the potential as oral an vaccine for *S. cerevisiae* expressing the capsid protein VP2 of IBDV. We constructed the recombinant *S. cerevisiae,* demonstrated that VP2 was displayed on the cell surface and had high immunoreactivity. By using the live ST1814G/Aga2-VP2 strain to immunize the mice, the results showed that recombinant *S. cerevisiae* significantly increased specific IgG and sIgA antibody titers, indicating the potential efficacy of vaccine-induced protection. These results suggested that the VP2 protein-expressing recombinant *S. cerevisiae* strain was a promising candidate oral subunit vaccine to prevent IBDV infection.

## 1. Introduction

IBDV, one of the *Birnaviridae* family, is a non-enveloped, double-stranded (ds) RNA virus, which can cause infectious bursal disease (IBD) [1]. This disease mainly occurs in 3–6 weeks chickens, characterized by the bursa of Fabricius (BF) atrophy. IBDV inhibits B cell response of acquired immunity through the destruction of lymphocytes in the BF, which increases susceptibility to other diseases and the probability of vaccination failure [2,3]. IBDV has strong resistance to many disinfectants. It is difficult to remove from the polluted places. Vaccination is still the only viable option to prevent IBD. At present, the traditional live attenuated vaccine and inactivated vaccine are still widely used against IBDV. But, given that the characteristics of IBDV vary, the potential risks of incomplete inactivation, and recovery of inactivated vaccine virulence, traditional vaccines cannot effectively protect against IBDV [4,5]. The subunit vaccine, with the advantages of good stability, high safety and purity, was one efficient strategy for vaccination of IBDV. The genome of IBDV consists of two dsRNA segments. The smaller segment encodes the RNA polymerase VP1 that is in charge of the genome’s replication, transcription, and virus packaging [4,6]. The larger segment encodes VP2, VP3, VP4, and VP5 [7,8]. VP2 is the major structural protein constituting the viral capsid, accounting for about 51% of the total viral proteins. As the major host-protective immunogen of IBDV, VP2 has more than three independent epitopes that can induce virus-neutralizing antibodies [9,10].

Various studies have described how VP2 was used as the excellent immunogen candidate of subunit vaccines to provide immune protection against IBDV [11]. Different heterologous protein expression systems, including the poxvirus [12], baculovirus [13], *Pichia pastoris* [14], and *Escherichia coli* system [15,16], have been used to express VP2 successfully. Recombinant turkey herpesvirus (rHVT)-IBDV vaccines expressing VP2 could be commercially available for application in chickens against IBDV. The routes of injection are needed after extensive purification for these systems, which are elaborate procedures. Mucosal immunity is the first line of defense system, and oral vaccines can evoke the protective mucosal immune response to infectious viruses through the oral route [17]. VP2 of IBDV production in different transgenic plant species, such as *Arabidopsis thaliana*, *Nicotiana benthamiana*, and *Oryza sativa*, has been reported in recent years. Specific antibodies were found in chickens after direct oral immunization of these plants without purification [18,19]. However, the recombinant VP2 production is the longsome procedure in transgenic plants. It is necessary to develop a safe, simple, and inexpensive oral vaccine delivery system.

*Saccharomyces cerevisiae* is a eukaryote host engineered to efficiently express various heterologous proteins. It has the characteristics of a clear genetic background and yeast expression system and is relatively stable and suitable for large-scale fermentation. Previous studies have shown that *S. cerevisiae* and its byproducts may have beneficial effects on immune efficacy and animal growth. The yeast cell could be phagocytosed by antigen-presenting cells, and some compositions, such as mannan and polysaccharides beta-1,3-D-glucan (BGs) located on yeast cell surface, could stimulate the immune response of body. The availability of *S. cerevisiae* expression system provides the basis for enabling ideal oral vaccine feasibility, design and development [20]. At present, some studies, such as porcine circovirus capsid protein recombinant *S. cerevisiae* [21], Dengue virus (DENV) surface display recombinant *S. cerevisiae* [22] and Japanese encephalitis virus (JEV) envelop protein surface display recombinant *S. cerevisiae* [23], have demonstrated that *S. cerevisiae* could be used as an effective oral vaccine system to express a variety of viral immunogenic antigens. Due to the Microfold (M) cells in the small intestine readily recognizing and presenting antigens, antigen presentation on the surface of yeast is more efficient than on the other systems [24]. However, the immune efficacy of *S. cerevisiae* surface displaying VP2 of IBDV still unclear.

In this study, we constructed a genome-integrated recombinant *S. cerevisiae* ST1814G/Aga2-VP2 expressing VP2 of IBDV using the GPI cell-surface display system of a-agglutinin. The results showed that mice orally administered the live ST1814G/Aga2-VP2 could induce certain immune protective response with IgG and secretory IgA antibodies. These findings suggested that the ST1814G/Aga2-VP2 may be a more economical, safer, and faster oral vaccine candidate strategy for preventing IBD.

## 2. Materials and Methods

### 2.1. Yeast Strains and Plasmids

The *S. cerevisiae* strain ST1814G (MATa aga1 his3Δ200 leu2Δ0 lys2Δ0 trp1Δ63 ura3Δ0 met15Δ0) and auxiliary plasmids were all kindly provided by Prof. Junbiao Dai [25]. The plasmid POT-TU (containing 667 bp GAPDH (Glyceraldehyde 3-phosphate dehydrogenase) promoter sequences, 207 bp Aga2 anchor sequences, and 187 bp ADH1 terminator sequences) was stored in our laboratory. Codon optimization, synthesis, and subcloning into the pMD19-T vector of VP2 gene were all outsourced (Genewiz Inc., Suzhou, China). The VP2 gene was subcloned into the yeast expression vector POT-TU by using a one-step cloning kit (Vazyme Biotech Co., Ltd., Nanjing, China). A 6 × His-tag is used at the C-terminal of VP2, which allows for tracking VP2 expression by the specific anti-His antibody, and an Aga2 signal peptide is used at the N terminal of VP2, which would grant the expression cassettes of recombinant proteins anchor on the surface of yeast. The plasmid after recombination was designated as pGPD-VP2-ADH1-POT.

### 2.2. Construction of Recombinant Yeast Strain Expressing VP2

The VP2 expression cassette was integrated into the yeast chromosome IV (Figure 1). Endonuclease *Bsm*b I (New England Biolabs (Beijing) Ltd., Beijing, China) was used to digest the plasmids of pMV-Leu2, pMV-URR1, and pMV-URR2, and *Bsa* I (New England Biolabs (Beijing) LTD., Beijing, China) was used to digest the pGPD-VP2-ADH1-POT. Then, the four fragments with specific cohesive ends after digestion were fused to one linearized fragment by T4 DNA ligase (New England Biolabs (Beijing) LTD., Beijing, China), which was transformed into ST1814G as previously described [26].

The individual transformants grown in SD-Leu auxotrophic medium were inoculated into 3 mL YPD medium and grown at 30 °C for 24 h with shaking 180 rpm. The extracted yeast genomic DNA was used as a template for PCR amplification analysis by the specific primer pair (F: 5′-GACGATAAGGTACCAGGATCCATGACAAACCTGCAAGATCA-3′ and R: 5′-GAATTCCACCACACTGGATCCCCTTATGGCCCGGATTATG-3′) to confirm the correct recombination of VP2 transcription units. In the recombinant yeast strain, the transcription level of VP2 was controlled by the GAPDH constitutive promoter.

### 2.3. Western Blotting Analysis

Mouse polyclonal antibody against recombinant VP2 that was expressed by the pET-28a system was prepared as previously described [27]. Rabbit anti-His monoclonal antibody (Yeasen Biotechnology (Shanghai) Co. Ltd., Shanghai, China) and HRP conjugated goat anti-rabbit IgG (Sungene Biotech Co. Ltd., Tianjin, China) were purchased for Western blotting.

The expression of recombinant VP2 in yeast was detected via Western blotting. After cultivation of the recombinant yeast in YPD liquid medium at 30 °C, yeast cells were harvested and lysed to gain the supernatant, which was performed for SDS-PAGE (12%) and Western Blot. The separated proteins were transferred to PVDF membrane (PALL, Carrolton, TX, USA), which was stained with rabbit anti-His-tag antibodies and incubated with HRP conjugated goat anti-rabbit IgG. Signals were developed with chemiluminescent substrate (Bio-Rad, Hercules, CA, USA), and images were acquired using the ChemiDoc Imaging System (Bio-Rad, USA). The VP2 expression was detected via Western blotting for three different recombinant ST1814G/Aga2-VP2 statins after cultivation for 24 h. Then, the high expression VP2 strain at 24 h was used to analyze the expression level of VP2 at the different cultivation time points of 24, 36, 48 and 60 h.

### 2.4. Quantitative Real-Time (qRT)-PCR

Total RNA was extracted using a Total RNA Extraction Kit (Yeasen Biotech, Shanghai, China) from the recombinant yeast. To obtain the first-strand cDNA using the 1st Strand cDNA Synthesis Kit (Yeasen Biotech, Shanghai, China). A 196-bp fragment of the *VP2* gene was amplified by the forward primer (5′-CTACACTATAACTGCAGCCGAT-3′) and reverse primer (5′-CGCAGTCCCATCAAAGCCTA-3′) in a total volume of 20 μL with a TransStart^®^ Top Green qPCR SuperMix. To detect the expression patterns of immune-related genes, the RNA was extracted from the spleen and thymus of mice, respectively (*n* = 3 per group). The primers were listed in Table 1, and relative expression of the genes was normalized to that of *β-actin*. Data were analyzed based on the 2^−ΔΔCT^ method. The balance of Th2/Th1 and Th17/Treg at the level of master transcription factors was determined by measuring the expression of effector T-cell master transcription factors.

### 2.5. Immunofluorescence Assay (IFA)

The location of VP2 on recombinant yeast cells was detected using IFA. The yeast cells were treated with 4% paraformaldehyde for 15 min and following incubation of rabbit anti-His-tag antibody overnight at 4 °C. The FITC-conjugated anti-rabbit IgG was incubated for an additional 1 h. Then, yeast cells were used for IFA using the confocal microscope (UltraVIEW VOX, PekinElmer, Waltham, MA, USA) as previously described [28].

### 2.6. Animal Immunization

The Laboratory Animal Ethical and Welfare Committee of Tianjin University (Permit number, TJUE-2021-051) provided the guidelines and regulations for the animal experiments.

First, 15 mice were randomly divided into 3 groups (*n* = 5 per group). To prepare the subunit vaccine candidate, the recombinant *S. cerevisiae* strain ST1814G/Aga2-VP2 was routinely cultured for 24 h with shaking at 180 rpm at 30 °C. The experimental protocol of group A was as follows: mice were orally administered with 100 μL PBS as a blank control and group B were fed with 1 × 10^9^ CFU ST1814G yeast fermentation liquid. Meanwhile, 1 × 10^9^ CFU ST1814G/Aga2-VP2 was taken orally per mouse in group C for immunization. Mice were boosted with an equal amount of the recombinant yeast cell liquid in the 1-week interval for a total of four immunizations and the whole experiment lasted 30 days. After the immunizations, all mice were euthanized. The spleen, thymus, and whole body of all mice were weighed to calculate the weight change and viscera index. The specific anti-IBDV antibody was detected via ELISA.

### 2.7. Enzyme-Linked Immunosorbent Assay (ELISA)

The specific IgG and sIgA were determined via ELISA in the serum and feces of mice, respectively, as previously described [28]. Briefly, the 96-well microplates were coated with purified VP2 [5 μg/mL (100 μL/well)] at 4 °C overnight. Then, wells were washed with PBST buffer (PBS containing 0.1% Tween-20) and blocked using 5% (*w/v*) BSA at 37 °C for 1 h. The 100 μL 1:100 diluted serum was added to each well for incubation. After washing with PBST, 100 μL HRP conjugated goat anti-mouse IgG or IgA (Abcam, Waltham, MA, USA) conjugator was added at 1:5000 dilutions, respectively. Then, the wells were incubated for 1 h at 37 °C. Afterward, the plates were washed with PBST buffer three times. TMB substrate solution was added to each well (100 μL/well). Following incubation in the dark at 37 °C for 15 min, the reaction was blocked by terminated liquid (2 M H_2_SO_4_) and the absorbance (450 nm) was recorded. The levels of antigen-specific IgG and sIgA were detected by calculating the positive/negative (P/N) value.

### 2.8. Statistical Analysis

All results are shown as the mean ± SD (standard deviations) with no less than three independent experiments. The experimental data were subjected to one-way analysis of variance (ANOVA) using the Graphpad Prism 7.0 software. The *p*-value was calculated using Student’s *t*-test. *p* < 0.05 was considered to be statistically significant.

## 3. Results

### 3.1. Construction of the Recombinant S. cerevisiae Strain ST1814G/Aga2-VP2

The IBDV-VP2 cDNA was PCR amplified from pMD19T-VP2 and transferred to the pGPD-ADH1-POT vector with a C-terminal 6 × His tag sequence (Figure 1 and Figure 2A). After the vectors pMV-Leu2, pMV-URR1, pMV-URR2, and pGPD-VP2-ADH1-POT were digested by endonuclease *Bsm*B I and *Bsa* I, respectively, the four fragments were fused by T4 DNA ligase and the fusion linearized fragment was introduced into *S. cerevisiae* strain ST1814G to achieve genomic integration (ST1814G/Aga2-VP2) (Figure 1 and Figure 2B). The results of agarose gel electrophoresis showed that the lengths of amplified pGPD-VP2-ADH1-POT vectors and ST1814G/Aga2-VP2 genome were ~1400 bp (Figure 2A,B). Then, the Western blotting analysis with anti-His-tag antibody confirmed the presence of an approximate 70 kDa VP2 band which is larger than the expected 48.3 kDa (Figure 2C) in three different recombinant strains ST1814G/Aga2-VP2. It is speculated that the VP2 protein may be glycosylated in yeast cells. The recombinant strain one (Lane 1) was chosen for other studies here, due to its high expression of VP2. The fluorescence was detected on the surface of recombinant *S. cerevisiae* (Figure 2D) with IFA using the confocal microscopy. The results indicated that VP2 was displayed on the surface of ST1814G/Aga2-VP2

### 3.2. The Fermentation Kinetics of Recombinant ST1814G/Aga2-VP2 Strain

In order to explore whether VP2 affects the growth characteristics of recombinant *S. cerevisiae*, the growth curves of ST1814G and ST1814G/Aga2-VP2 were measured. They both entered the logarithmic growth and stable phase within a relatively synchronous time. Compared with ST1814G, the growth characteristics of recombinant strains ST1814G/Aga2-VP2 did not show a significant discrepancy (Figure 3A). The cell amount of both yeasts reached the peak at 36 h and showed growth stagnation after 36 h (Figure 3B). These results showed that the expression of VP2 did not significantly affect the growth of *S. cerevisiae* strain ST1814G.

To investigate the optimal time for the expression of recombinant VP2 in ST1814G/Aga2-VP2, the expression level of VP2 gene and VP2 protein at 12, 24, 36, 48 and 60 h were detected via qRT-PCR and Western blotting analysis, respectively. As shown in Figure 3C, the transcription level of VP2 reached the peak level at 24 h and then decreased slowly until 60 h. The accumulation level of VP2 protein (~70 kDa) was detected at 24, 36, 48 and 60 h and peaked at 24 h in recombinant strain ST1814G/Aga2-VP2, but not in ST1814G (Figure 3D).

### 3.3. Health Status of the Mice Post Oral Immunization

The weight gain and viscera index could be used as an important health index. The weight of mice was measured in different periods after oral immunization. No significant difference was observed among the various groups in weight gain (Figure 4A). This indicated that the recombinant ST1814G/Aga2-VP2 did not interfere with the normal growth of mice. The viscera indexes of spleen and thymus were compared on day 23 after primary immunization among the different groups. Compared with the PBS and ST1814G groups, an increase was observed for the viscera index of spleen and thymus in the mice post oral live ST1814G/Aga2-VP2 (Figure 4B). The results indicated that the recombinant *S. cerevisiae* strain ST1814G/Aga2-VP2 could induce the humoral immune response.

### 3.4. Antibody Levels of IgG and sIgA Post Oral Immunization

The immunogenicity of recombinant *S. cerevisiae* strain ST1814G/Aga2-VP2 was analyzed by measuring the levels of VP2-specific IgG and sIgA antibodies in mice after oral immunization. These antibodies in serum and feces were detected via indirect ELISA to assess the systemic and mucosal immune response. At the time of immunization, there were no significant changes in the levels of IgG and sIgA (Figure 5). As shown in Figure 5A, a signification rise of specific IgG in serum was observed after oral immunization with ST1814G/Aga2-VP2. And the antibody level was significantly higher than that treated with ST1814G. Moreover, the sIgA antibody levels in the feces of mice treated with ST1814G/Aga2-VP2 increased after the oral immunization. And significantly higher sIgA titers were detected in the mice treated with ST1814G/Aga2-VP2 than those with ST1814G (Figure 5B). These results indicated that the recombinant live ST1814G/Aga2-VP2 could significantly induce specific IgG and sIgA antibodies in mice after oral immunization.

### 3.5. The Expression Patterns of Cytokines in Spleen and Thymus of the Immunized Mice

To analyze the production of cytokines induced by ST1814G/Aga2-VP2 in mice after oral immunization, the mRNA levels of cytokines in spleen and thymus were detected via qRT-PCR. Significantly higher levels of the Th1-associated cytokines IFN-γ, IL-2 and anti-inflammatory cytokines IL-10 in spleen were induced by ST1814G/Aga2-VP2 than those by PBS and ST1814G. The transcript levels of IFN-α in the thymus of ST1814G/Aga2-VP2 group were higher than these of PBS or ST1814G groups (Figure 6A). The TGF-β mRNA level in spleen with ST1814G/Aga2-VP2 trended higher than that with PBS or ST1814G, but this difference failed to reach statistical significance. Notably, the TGF-β mRNA level of the thymus in the ST1814G/Aga2-VP2 group was higher than that in ST1814G and PBS group. The transcriptional levels of TLR3, IDO and MyD88 in the ST1814G/Aga2-VP2 group were significantly higher than those in the PBS group (Figure 6B). These results showed that ST1814G/Aga2-VP2 could increase the expression of IFN-α, IFN-γ, IL-2, IL-10, TLR3, IDO, MyD88 and TGF-β at the transcriptional level.

To investigate the effect of ST1814G/Aga2-VP2 on T-cell differentiation, the transcriptional levels of GATA3, RORC, TBX21 and Foxp3 in spleen and thymus were determined via qRT-PCR. The mRNA levels of TBX21, GATA3 and Foxp3 in the ST1814G/Aga2-VP2 group were significantly higher than that in ST1814G groups. The mRNA level of RORC was no significantly difference among the groups (Figure 6C). The results indicated that ST1814G/Aga2-VP2 could promote the differentiation of various T cells and improve the immune regulation ability. The ratio of Th2/Th1 and Th17/Treg cells in the recombinant yeast group was significantly lower than in the control group, which demonstrated ST1814G/Aga2-VP2 could increase the proportion of Th1 and Treg cells (Figure 6D). These results showed that a recombinant *S. cerevisiae* strain ST1814G/Aga2-VP2 could stimulate the immune system to regulate the body’s immune balance.

## 4. Discussion

IBDV, the pathogen of Gumboro disease, causes significant morbidity and mortality [29]. To date, the main strategy to control IBDV is still immunity vaccination with inactivated or live attenuated vaccines [30,31]. Due to the advantages of high reliability and safety, the subunit vaccine has been developed against IBDV. The VP2, as the main protein that forms the viral capsid, is the ideal candidate for the IBDV subunit vaccine. Various expression systems, such as insect cells [32], mammalian cells [33], transgenic plants [34] and *E. coli* [35], have been used to express VP2 for the preparation of a subunit vaccine. HVT, originally isolated from domestic turkeys, has been used as a recombinant vector for the IBDV vaccine on the market. The HVT-IBDV vaccines confer long-lasting protective immunity against pathogens in chickens. However, most of the systems need to achieve purification or long-time growth of VP2. Compared with subcutaneous injection of VP2 or viral capsid, the oral immunization could effectively stimulate protective mucosal and systemic immune response, and the process is simple, non-invasive, inexpensive and convenient. So, one safe and effective oral vaccine strategy is needed to prevent IBDV. Compared with other oral platforms, *S. cerevisiae*, as a delivery vehicle for oral vaccines, has several inherent advantages. They have been used in the fermentation of drink and food with GRAS status for thousands of years [36]. Furthermore, the yeast can remain for a relatively long time to ensure enough time is allocated for it to be processed and absorbed in the gut [37]. In addition, *S. cerevisiae* could stimulate immunity and maintain intestinal homeostasis [38]. They can be used as a natural immune adjuvant to improve the immune efficacy of oral vaccines and enhance the immunogenicity of antigens [39,40]. Therefore, in this study, *S. cerevisiae* were used as the host strain to express VP2 of IBDV.

Plasmids are considered a burden during host passaging. To avoid the loss of plasmids in the yeast large-scale fermentation process without antibiotic selection, the expression cassette of VP2 was integrated into the Chr IV of ST1814G via homologous recombination (Figure 1). In addition, considering that the antigen surface-displayed *S. cerevisiae* could better enable antigen recognition and presentation by M cells in the small intestine, the lectin GPI surface display expression system was used to display VP2 protein on the surface of yeast. However, in this study, the molecular weight of VP2 was larger (~70 kDa) than expected (~48.3 kDa), which may be related to N-glycosylation during the post-translational modification in yeast. A previous study using yeast to express Fiber-2 of FAdV-4 also found that the recombinant protein was enlarged by glycosylation [28]. Several studies drew different conclusions on whether post-translational N-glycosylation in *S. cerevisiae* has an effect on the recombinant protein. Some studies showed that the presence and number of mannose had an effect on the immunogenicity and oligomerization of some antigens, and that glycosylation enhanced the thermostability of recombinant proteins [41]. However, some studies have pointed out that excessive glycosylation could reduce the expression of recombinant proteins [42], and excessive glycosylation of the specific parts of antigen may reduce immunogenicity. The post-translational N-glycosylation modification of proteins in yeast cells has a complex mechanism and process. Whether it has an effect on the function of VP2 protein needs to be further studied.

There was no significant difference in the body weight gain among each group after the immunization. The organ index of thymus and spleen showed no significant difference for the groups (Figure 4). These results prove that the constructed recombinant ST1814G/Aga2-VP2 strain has good safety and does not create an additional burden for the body’s growth.

The sIgA in mucosa and IgG in serum could well reflect the strength of the humoral immune response [43]. In ST1814G/Aga2-VP2 group, the levels of IgG and sIgA were significantly higher than those in the control group (Figure 5). A previous study has shown that in chicken, IBDV is rich in lymphoid tissue and is easily transferred to the blood [44]. Then, the virus infected peripheral blood mononuclear cells that were transferred to the bursa of Fabricius via the blood circulation [44]. Due to the high level of IgG antibodies in the blood, IBDV is more likely to combine with IgG to form a complex and be phagocytosed by macrophages, while sIgA antibodies in the mucosa can prevent IBDV from adhering to the bursa, reducing the viral load. Therefore, oral administration of the recombinant ST1814/Aga2-VP2 could induce a strong humoral immune response to anti-IBDV infection.

TLR3, as a Toll-like receptor, mainly recognizes viral RNA in endosomes in innate immunity. The activated Toll interleukin-1 receptor (TIR) domain recruits several adaptors, including TRIF and MyD88 [45]. In this study, the recombinant ST1814G/Aga2-VP2 elicited higher levels of TLR3 and MyD88 (Figure 6B), proving that the vaccine candidate we constructed here could better stimulate the NF-κB, IRFs and AP-1e signaling pathways and promote IFN production. IFN-1 and IFN-γ can be induced by the NF-κB signaling pathway and AP-1 signaling pathway, respectively. A large number of evidence shows that IFN could inhibit the replication of avian influenza epidemic viruses, such as avian influenza viruses, IBDV, infectious bronchitis virus, and Newcastle disease virus [46,47,48]. Compared with the control group, the group with ST1814G/Aga2-VP2 had higher levels of IFN-α and IFN-γ, indicating that the recombinant yeast could improve the body’s antiviral and immune regulation ability (Figure 6A).

CD4^+^ T cells play a key role in both innate and adaptive immunity [49,50]. The naive conventional CD4 T cells mainly differentiate into several distinct CD4^+^ T cells, such as Th1, Th2, Th17 and induced regulatory T (Treg) cells. Th1 cells mediate immune responses to intracellular pathogens, and their major cytokine products are IFN-γ, lymphotoxin alpha (LTα) and IL-2. Th2 cells are mainly responsible for humoral immunity. They secrete lymphokines such as interleukins, and the IL-4 can promote the proliferation of B cells and induce the production of antibodies, especially the production of IgE [51,52]. In this study, the representative transcription factors Tbx21 and Gata of Th1 and Th2 in the ST1814G/Aga2-VP2 group were significantly up-regulated (Figure 6C), indicating that the recombinant yeast immune preparation may stimulate cellular and humoral immune responses. Th17 cells play an important role in immune responses against extracellular bacteria and fungi [53]. Treg cells could maintain self-tolerance and regulate immune responses, with immunosuppressive and immune homeostasis functions [54]. In this study, the mRNA expression level of RORC and the representative transcription factor of Th17 was decreased, while Foxp3, the transcription factor of Treg, was significantly increased. These results showed that the recombinant ST1814G/Aga2-VP2 may improve the body’s ability to maintain immune homeostasis (Figure 6C). However, there is no Foxp3 homologous gene in chickens. The results about this transcription factor here may not be suitable for chickens. In addition, we also observed up-regulation of the transcriptional level of inhibitory cytokine IL-10 and immune tolerance indicator IDO. Considering that IL-10 can inhibit the production of IL-23 and avoid excessive immune response [55], and IDO can promote Treg cells differentiation [56], these data suggested that recombinant yeast could have potential for maintaining intestinal immune tolerance and mucosal homeostasis.

In the body’s normal state, Th1/Th2 and Th17/Treg are in a state of balance. After oral immune preparation, the balance of Th1/Th2 is tilted to Th1, to increase the expression of interferon and enhance the body’s immunity. The Th17/Treg balance is tilted towards Treg to avoid histiocyte infiltration and tissue destruction caused by the overexpression of Th17-related cytokines (Figure 6D). This indicates that the recombinant ST1814G/Aga2-VP2 constructed in this study has potential protective efficacy and certain safety for further development of IBDV subunit vaccine. However, if the balance of Th1/Th2 and Th17/Treg cell could be detected in chickens after oral immunization via flow cytometry, the data might be more convincing. Moreover, IBDV mainly causes immunosuppression in chickens. The animal experiments with chickens were needed to accurately confirm the effect of oral vaccines.

## 5. Conclusions

In this study, we successfully constructed a recombinant *S. cerevisiae* strain ST1814G/Aga2-VP2 that displayed the VP2 of IBDV on the cell surface. The live ST1814G/Aga2-VP2 could induce specific IgG and sIgA antibodies and increase the expression levels of immune-related transcription factors in mice. These results suggested that oral live recombinant *S. cerevisiae* ST1814G/Aga2-VP2 is a potentially safe and effective strategy for controlling IBDV infection.

## Figures and Tables

**Figure 1 vaccines-11-01849-f001:**
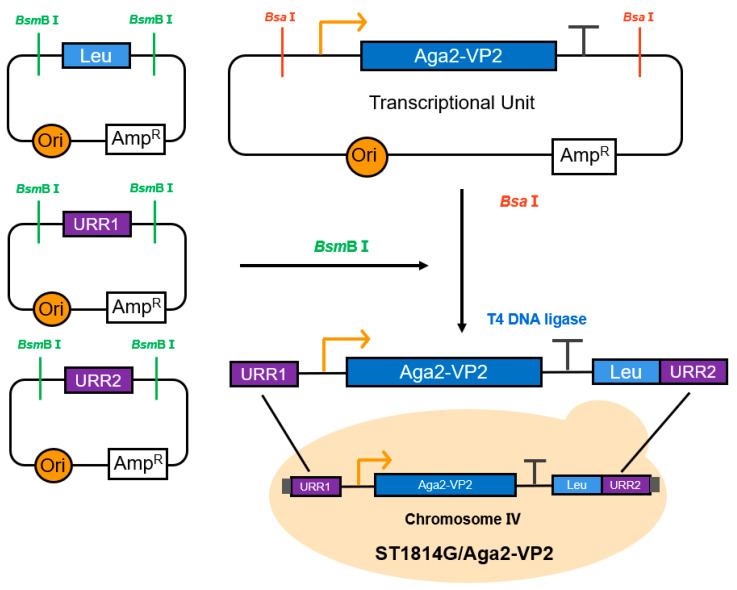
Workflow diagram for constructing recombinant yeast. *Bsm*B I was used to digest pMV-Leu2, pMV-URR1, and pMV-URR2, and *Bsa* I was used to digest pGPD-VP2-ADH1-POT. Then, four fragments of enzyme digestion were fused sequentially to one fragment by T4 DNA ligase, which was transformed into yeast strain ST1814G. The recombinant yeast strain could express VP2 by controlling GAPDH promoter and produce LEU by itself growing on SD-Leu auxotrophic medium.

**Figure 2 vaccines-11-01849-f002:**
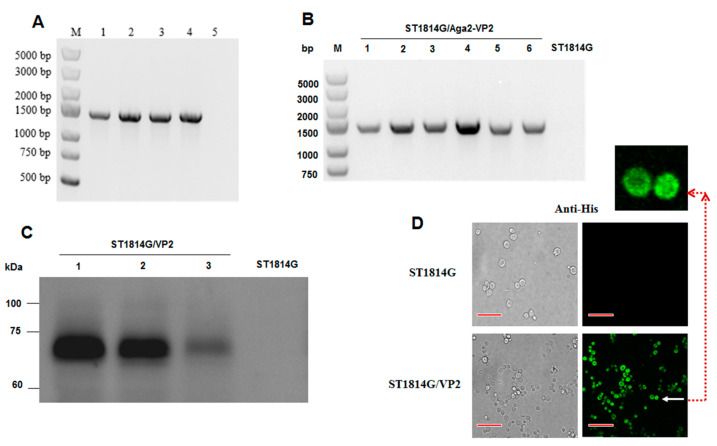
Construction and identification of the recombinant *S. cerevisiae* strain expressing VP2. (**A**) PCR amplification of pGPD-VP2-ADH1-POT (1359 bp, lane 1–4: the vectors of pGPD-VP2-ADH1-POT extracted from four random *E. coli* clones were used as templates, lane 5: the vector of POT-GFP as negtive control). (**B**) VP2 fragments amplified by genomic PCR to screen positive yeast clones. (1359 bp, lanes 1–6: the genomic DNA extracting from six randomly selected yeast clones was used for PCR) (**C**) Western blot detection of VP2 expression in recombinant *S*. *cerevisiae* strains (70 kDa, lanes 1–3: VP2 expression in three randomly selected recombinant yeast strains). (**D**) Detection of VP2 expressed on the surface of recombinant yeast strain by IFA using the confocal microscopy (the bar = 100 μm).

**Figure 3 vaccines-11-01849-f003:**
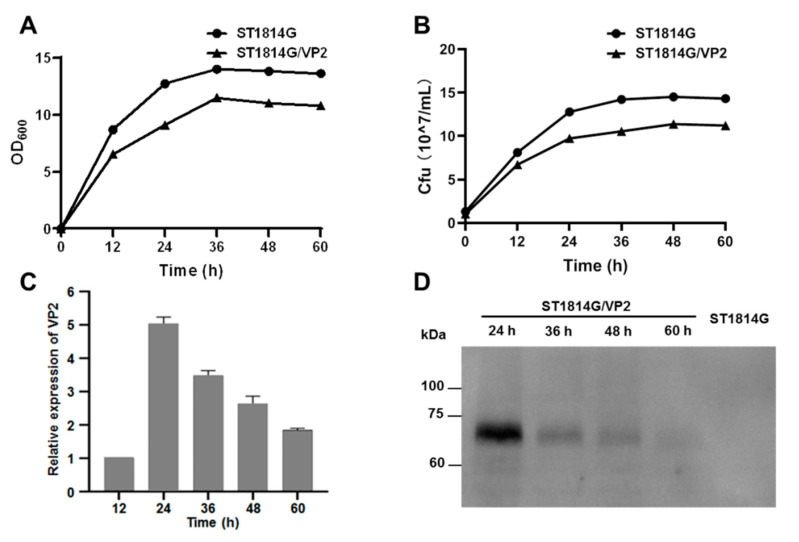
The fermentation and VP2 expression kinetic of ST1814G/VP2. (**A**) The growth curve of ST1814G and ST1814G/VP2. The OD_600_ value was determined at the time points of 12, 24, 36, 48, and 60 h. (**B**) The number of ST1814G and ST1814G/VP2 cells was measured via plate counting at the same time. (**C**) The transcriptional expression of VP2 at the cultivation time points of 12, 24, 36, 48, and 60 h was detected via qRT-PCR. (**D**) The expression of VP2 in ST1814G/VP2 strain was detected via Western blotting at the cultivation time points of 24, 36, 48, and 60 h. ST1814G was used as negative control.

**Figure 4 vaccines-11-01849-f004:**
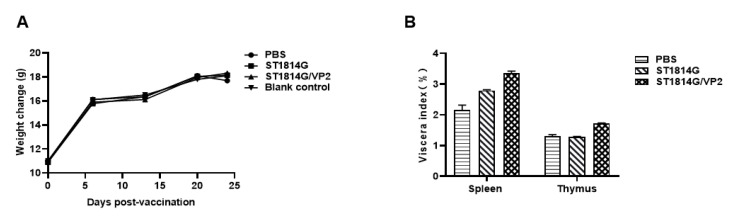
Effects of subunit vaccine candidate on mice growth status. (**A**) The body weight change curve was drawn to record at a 5-day interval during the experiment (three random mice from each group). (**B**) The ratio of wet weight/body weight of the spleen and thymus of mice was calculated after the immunizations. The data were obtained from three independent experiments and represent the means ± SD.

**Figure 5 vaccines-11-01849-f005:**
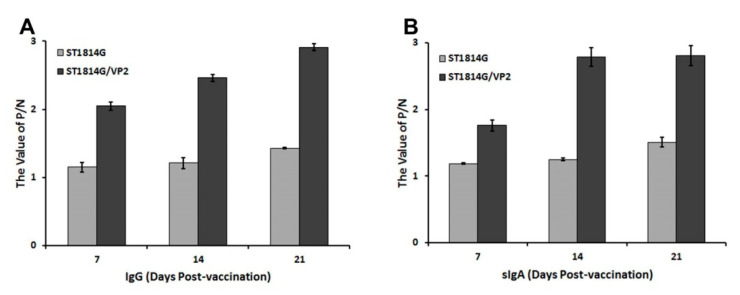
Kinetics of VP2-specific IgG in serum and sIgA in feces of vaccinated mice. The levels of VP2-specific IgG in serum (**A**) and sIgA in feces (**B**) of vaccinated mice were determined, and the differences were compared among groups (*n* = 3 per group), respectively. The data here were obtained from three independent experiments and represent the means ± SD.

**Figure 6 vaccines-11-01849-f006:**
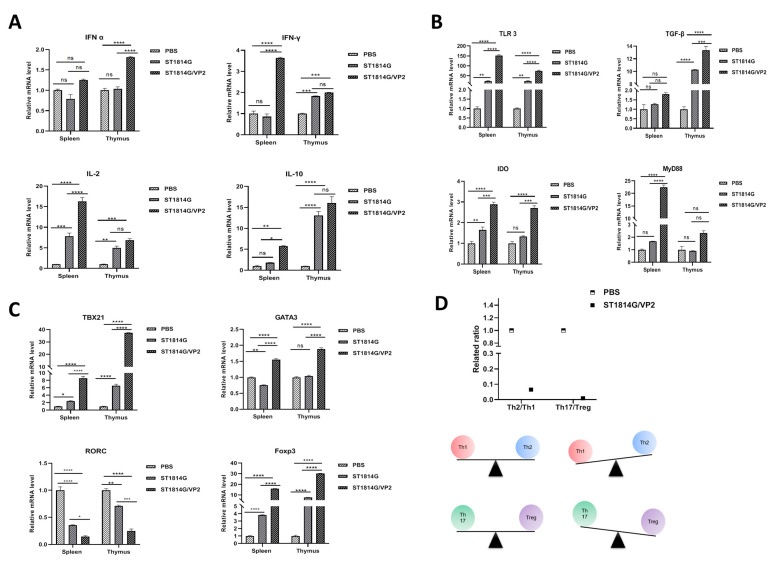
The mRNA expression levels of various immune-related cytokines in spleen and thymus. The transcriptional levels of IFN-α, IFN-β, IL-2, IL-10 (**A**), TLR-3, TGF-β, IDO, MyD88 (**B**), TBX21, Gata3, RORC and Foxp3 (**C**) of the immunized mice (*n* = 3 per group) were detected via qRT-PCR. The relative mRNA expression was normalized to that of β-actin. The data were obtained from three independent experiments, calculated based on the 2^−ΔΔCT^ method, and represent the averages ± SD. The significance of differences was determined via two-way analysis of variance (ns, no significance; *, *p* < 0.05; **, *p* < 0.01; ***, *p* < 0.001 or ****, *p* < 0.0001). (**D**) The ratio of Th2/Th1 and Th17/Treg was determined after oral immunization and the change in immune balance was analyzed based on the expression of effector T-cell master transcription factors.

**Table 1 vaccines-11-01849-t001:** Primers used in qRT-PCR.

Primer Name	Genbank Accession Number	Primer Sequence
β-actin-F	NM_007393.5	AGAGGGAAATCGTGCGTGAC
β-actin-R		CAATAGTGATGACCTGGCCGT
VP2-F1	KT381974.1	CTACACTATAACTGCAGCCGAT
VP2-R1		CGCAGTCCCATCAAAGCCTA
IL-2-F	NM_008366.3	GTGCTCCTTGTCAACAGCG
IL-2-R		GGGGAGTTTCAGGTTCCTGTA
IL-10-F	NM_010548.2	GGTTGCCAAGCCTTATCGGA
IL-10-R		ACCTGCTCCACTGCCTTGCT
IFN-α-F	NM_010503.2	CTAGACTCTGTGCTTTCCTCGT
IFN-α-R		ATCGCATCCTAGAGAACAGGT
IFN-γ-F	NM_008337.4	ATGAACGCTACACACTGCATC
IFN-γ-R		CCATCCTTTTGCCAGTTCCTC
MyD88-F	NM_010851.3	GAGATGATCCGGCAACTAGAAC
MyD88-R		GTCCTTCTTCATCGCCTTGTAT
TGF-β-F	NM_011577.2	CTCCCGTGGCTTCTAGTGC
TGF-β-R		GCCTTAGTTTGGACAGGATCTG
TLR3-F	NM_001357316.1	AGTACAACAATATACAGCGTCT
TLR3-R		TGCTTAGTAAATGCTCGCTTC
Gata3-F	NM_008091.3	CTGGAGGAGGAACGCTAATG
Gata3-R		GATGACATGTGTCTGGAGAGAG
Foxp3-F	NM_001199347.1	GTGGCCTCAATGGACAAGA
Foxp3-R		AAGGGTGGCATAGGTGAAAG
TBX21-F	NM_019507.2	CACATCGTGGAGGTGAATGA
TBX21-R		CTTCTCACCTCTTCTATCCAACC
RORC-F	NM_001293734.1	CGGAGCAGACACACTTACATAC
RORC-R		CTTTGCCTCGTTCTGGACTATAC
IDO-F	M69109.1	CCTGCCTCCTATTCTGTCTTATG
IDO-R		ACATAGCTACCAGTCTGGAGA

## Data Availability

The data that support this study could be available on reasonable request from the corresponding author (J.H. or L.Z.).

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
