# Peer review of "Oral Immunization with Recombinant Saccharomyces cerevisiae Expressing Viral Capsid Protein 2 of Infectious Bursal Disease Virus Induces Unique Specific Antibodies and Protective Immunity"

_vaccines, 2023, doi:10.3390/vaccines11121849_

Round 1
Reviewer 1 Report (Previous Reviewer 2)
Comments and Suggestions for Authors
General comments
The conclusions in the present paper are over-interpreted. In addition, the required data are not provided. The authors need to correctly state the conclusions and interpretations that can be drawn from the objectives, methods and results of this study, to avoid misleading the readers.
Specific comments
The authors need to show the presence of antibodies specific for IBDV-VP2 by western blotting using serum from immunized mice. The objectives of this study is to investigate the potential of this oral vaccine. Therefore, the authors need to show the evidence of the antibody production that can specifically recognize with IBDV-VP2.
In addition, importantly, the authors need to state the limitations of this study and should add some discussions as following contents;
1) To accurately know the effect of oral vaccines in the present study, it has to be determined by the animal experiments using chickens.
2) The expression levels of transcriptional factors may show the increase or decrease in the proportion of T cells subtypes; however, to accurately know T cell dynamics in immunized animals, the analyses should be conducted by flow cytometry.
3) Chickens lack the Foxp3 gene. Therefore, the data obtained from the animal experiments using mice, especially Treg cells, can not translate well to chickens. To accurately know the alterations in T cell dynamics induced by the immunization with oral vaccines in the present study, the analyses using immunized chickens are required.
Other comments
Line 14 objective to aims
Lines 20-21 diversity of intestinal flora
Line 21 stimulate
Lines 22-23 Correct grammatical errors.
Line 49 turkey herpesvirus
Lines 50-51 injection immunization?
Line 79 showed that
Line 80 secretory IgA (sIgA)
Line 213 strains but not clones?
Line 216 randow?
Lines 267-268 the significantly higher flow cytometryIgA? This sentence has to be amended.
Lines 269-271 If the authors conclude as described here, neutralization assays using the virus have to be conducted. These results alone do not indicate neutralizing capacity.
Line 263 shown
Line 331 the VP2
Line 371 prove
Lines 376-378 The authors did not provide any data to show the production of antibodies specific to IBDV-VP2 and their neutralization activity. Thus, this discussion is an over-interpretation.
Line 378 A previous study has shown
Lines 378-379 no reference
Line 380 and?
Line 423 rmaster?
Line 442 The authors did not provide any data showing the production of specific antibodies.
Comments on the Quality of English LanguageThere are still some grammatical and careless mistakes. A proofreading by a native speaker is required.
Author Response
Dear Reviewer,
We appreciate very much for your professional review work on our manuscript. In accordance with the valuable suggestions and comments, we have made extensive corrections to our previous draft, and the detailed corrections are highlighted in the manuscript. Point-by-point responses to the the Reviewer are listed below in the attachment.

Reviewer 2 Report (New Reviewer)
Comments and Suggestions for Authors
The draft in general is OK, but need minor revision before it can be accepted for publication in MDPI Vaccines. My comments and suggestions to authors are given below
1- Draft need minor english editing
2-In figure 2, image A and B can be inverted so that bands appear black on white background.
3-in Figure 2, image D, author showed zoom in one cell and show it in separate panel so that expressed protein can be seen on cell surface properly
4- Again in figure 2, image D, value of scale bar is missing
5-Since yeast display is essentially like whole recombinant yeast, author should discuss the advantage of this platform over other like purified protein immunogen in terms of stability, shelf life , storage temp. Check recent relevant paper on yeast based vaccines.
6-Discussion section is quite lengthy, just discuss need information
7-Why there is VP2 specific IgG and sIgA in yeast control only????
Comments on the Quality of English Language
Overall english is OK
Author Response
Dear Reviewer,
We appreciate very much for your professional review work and positive comments on our manuscript. In accordance with the valuable suggestions and comments, we have made extensive corrections to our previous draft, and the detailed corrections are highlighted in the manuscript. Point-by-point responses to you are listed in the attachment.
Reviewer 2:
Comments and Suggestions for Authors
The draft in general is OK, but need minor revision before it can be accepted for publication in MDPI Vaccines. My comments and suggestions to authors are given below
-Response: Thank you very much for your professional and positive comments on our draft. In accordance with your valuable suggestions, we have made extensive corrections to our previous draft, and the detailed corrections are listed below.
1- Draft need minor english editing
-Response: Thank you for reading the manuscript carefully. We have rewritten and proofread the manuscript carefully
2-In figure 2, image A and B can be inverted so that bands appear black on white background.
-Response: Thank you for this suggestion. We have inverted the Fig. 2A and 2B.
3-in Figure 2, image D, author showed zoom in one cell and show it in separate panel so that expressed protein can be seen on cell surface properly.
-Response: Thank you for this valuable suggestion. We have shown two yeast cells in separate panel in Figure 2.
4- Again in figure 2, image D, value of scale bar is missing
-Response: Sorry for the error of Figures 2D, and we have added the value of scale bar in Figure 2 legend.
5-Since yeast display is essentially like whole recombinant yeast, author should discuss the advantage of this platform over other like purified protein immunogen in terms of stability, shelf life , storage temp. Check recent relevant paper on yeast based vaccines.
-Response: Thank you very much for the valuable suggestions. We added the advantages of the yeast platform in Discussion.
6-Discussion section is quite lengthy, just discuss need information
-Response: We have rewrittern the section of “Discussion”. Thank you very much.
7-Why there is VP2 specific IgG and sIgA in yeast control only????
-Response: Thank you very much for this valuable question. We measured the levels of VP2 sepcific IgG and sIgA in mice by ELISA. Although we could found there may be IgG and sIgA in yeast ST1814G group, the levels was low and did not increase after oral immunization. So we suppose these may be the non-sepcificity.
Comments on the Quality of English Language:Overall english is OK
-Response: Thank you for your positive comments. We have rewritten the manuscript carefully.
Round 2
Reviewer 1 Report (Previous Reviewer 2)
Comments and Suggestions for Authors
The authors have toned down their statement. The conclusion seems reasonable.
Comments on the Quality of English LanguageThere seems to be minor grammatical errors.
Author Response
Dear Reviewer:
Thank you very much for your positive comments on our draft. We have revised and proofread the manuscript carefully to minimize the grammatical errors. The detailed changes are highlighted in the manuscript.
Thank you again.
This manuscript is a resubmission of an earlier submission. The following is a list of the peer review reports and author responses from that submission.
Round 1
Reviewer 1 Report
Comments and Suggestions for Authors
1. English must be improved; there are fundamental language issues that must be fixed before publication.
2. Lines 62-63. Are only the mannan and polysaccharides phagocytose or is the whole cell phagocytosed? This sentence is not clear at all.
3. Is it the expression of VP2 conditional or constitutive? Please explain in the methods and materials how the expression is controlled. This point is related to comment 5.
4. Figure 2D. All the cells are green. How do they know that the VP2 protein is at the cell membrane? The resolution of that micrographs does not allow us to determine so. The only way to do it by fluorescence microscopy would be by confocal microscopy.
5. Lines 228-229. If the VP2 gene was inserted into the chromosome, then why would the transcription of the VP2 RNA decease with time? Is this related to the expression system or to the stage of cellular growth? Is there a relationship between the cell growth (figures 2A and 2B with 2C)=
6. Figure 3C. The y-axis should indicate the relative expression of “What gene”?Also, why are the color of the bars different?
7. What is the weight percentage of the VP2 protein per gram of cell? This value is important to control the dose.
8. If the expression of VP2 decays with time (which I am not clear why) how do you control the dose of VP2? Is the yeast killed before immunization? If not, how can you be sure that yeast is expressing sufficient amounts of VP2?
9. Please explain why the expression decays if the gene of interest is inserted in a chromosome.
10. LIne 264 - why is there an interrogation sign?
11. Figures 5. There is something wrong here. These figures are supposed to determine IgG and IgA which are measured by ELISA. Nonetheless, the y-axis indicates Relative mRNA levels. Furthermore, Why are the IgG and IgA levels for both groups start at the same point?; there is no reason why in Figure 5A both curves have the same value at 0 days post-vaccination.
12. Section 3.4 How many mice per group were analyzed?
13. Figure 6. Please indicate in the y-axis that is the relative expression of the gene of interest with respect to which particular gene. Please indicate in the figure legend the method used to calculate the relative expression of the gene of interest. How many data points there are per bar? Was the per done in duplicate or triplicate? How many individuals per group were analyzed?
14. It is not clear why is the alteration of gut microbiota important to the development of vaccines. Please justify this part. In fact, figure 7 shows that the changes in the gut microbiota population are negligible; without a statistical analysis to show that the populations are different there is no evidence that the statement in lines 345-347 is true at all. I would suggest removing this section.
15. Lines 376-378. It is normal to expect that glycosylation would modify the molecular weight of a protein, but is the observed MW consistent with other expression systems for this protein? In fact, a more important question is: is VP2 glycosylated in the natural host? If not, would this affect its immunogenicity?
16. How does the immune response of this vaccine compare to that of the commercially available vaccines?
Comments on the Quality of English LanguageThe article must be completely re-written and supervised by a translator.
Reviewer 2 Report
Comments and Suggestions for Authors
General comments
The authors constructed Saccharomyces cerevisiae expressing IBDV-VP2 as a vaccine against IBD. The IBDV-VP2 was successfully expressed on the plasma membrane, and its oral immunization induced antibody production in mice. However, this manuscript did not provide the effect of this vaccine on chickens. In addition, the strategy of oral immunization for chickens was not described. It is difficult to envision the application of this vaccine for the protection against avian diseases. This manuscript includes scientifically incorrect methods and observations. Therefore, the authors need to reconsider the objectives and methods of the study.
Other comments
- This manuscript includes many grammatical errors. The authors need to carefully edit it.
- Lines 46-48: IBDV-HVT vaccines are commercially available. This vaccine can be used for the protection against Marek's disease and IBD. In the view of objectives of this study, the authors must mention about this vaccine. Also, advantages of oral immunization should be described compared with this vaccine, and the advantages over the use of the HVT-IBDV vaccine should be presented.
- Lines 77-78: Why did the authors choose the mouse model to evaluate the efficacy? IBDV is an avian disease. The reasons why the authors use mice for the assessment as a vaccine antigen have to be clearly stated.
- Line 80: secretory IgA (sIgA), and the authors need to unify whether it's “SIgA” or “sIgA”.
- Line 163: qRT-PCR? To assess the antibody titer, did the authors perform qRT-PCR?
- Line 212: What does each lane indicate in Figure 2 A, B and C? What are the difference in each lane?
- Line 264: “anti-IBDV? specific IgG antibody levels”? What would the authors like to say here? Also, the authors need to show the presence of antibodies specific for IBDV-VP2 by western blotting.
- Line 271: The authors solely observed the antibody production and did not observe their neutralizing abilities here.
- Lines 307-309: How did the authors analyze the ratio of T cell populations? The description regarding the flow cytometry analysis was not found in the materials and methods.
- Lines345-347: There seemed to be no difference in the diversity of commensal bacteria in the gut between PBS- and ST1814G/Aga2-VP2 groups. Why did the authors perform this analysis? Is there a relationship with the immune response induced by the immunization with ST1814G/Aga2-VP2?
Comments on the Quality of English LanguageThis manuscript includes many grammatical errors. The authors need to carefully edit it.